

# The outcomes of most aggressive interactions among closely related bird species are asymmetric

Paul R. Martin[1], Cameron Freshwater[2] and Cameron K. Ghalambor[3]

[1] Department of Biology, Queen's University, Kingston, Ontario, Canada
[2] Department of Biology, University of Victoria, Victoria, British Columbia, Canada
[3] Department of Biology and Graduate Degree Program in Ecology, Colorado State University, Fort Collins, Colorado, United States

## ABSTRACT

Aggressive interactions among closely related species are common, and can play an important role as a selective pressure shaping species traits and assemblages. The nature of this selective pressure depends on whether the outcomes of aggressive contests are asymmetric between species (i.e., one species is consistently dominant), yet few studies have estimated the prevalence of asymmetric versus symmetric outcomes to aggressive contests. Here we use previously published data involving 26,212 interactions between 270 species pairs of birds from 26 taxonomic families to address the question: How often are aggressive interactions among closely related bird species asymmetric? We define asymmetry using (i) the proportion of contests won by one species, and (ii) statistical tests for asymmetric outcomes of aggressive contests. We calculate these asymmetries using data summed across different sites for each species pair, and compare results to asymmetries calculated using data separated by location. We find that 80% of species pairs had aggressive outcomes where one species won 80% or more of aggressive contests. We also find that the majority of aggressive interactions among closely related species show statistically significant asymmetries, and above a sample size of 52 interactions, all outcomes are asymmetric following binomial tests. Species pairs with dominance data from multiple sites showed the same dominance relationship across locations in 93% of the species pairs. Overall, our results suggest that the outcome of aggressive interactions among closely related species are usually consistent and asymmetric, and should thus favor ecological and evolutionary strategies specific to the position of a species within a dominance hierarchy.

## INTRODUCTION

Aggressive interactions commonly occur among closely related species (*Kruuk, 1967*; *Feinsinger, 1976*; *Willis & Oniki, 1978*; *Robinson & Terborgh, 1995*). Such direct interspecific interactions have been shown to play an important role in interference competition for resources, including habitat, food, nest sites, and roost sites (*Chappell, 1978*; *Dhondt & Eyckerman, 1980*; *Robertson & Gaines, 1986*; *Alatalo & Moreno, 1987*; *Wallace, Collier*

Corresponding author
Paul R. Martin, pm45@queensu.ca

& Sydeman, 1992; Dhondt, 2012). Aggressive interactions may also reduce the costs of indirect ecological interactions (*Martin, 1988*; *Martin & Martin, 2001a*; *Martin & Martin, 2001b*) that are generally referred to as "apparent competition", such as density-dependent predation or parasitism involving multiple prey or host species (*Holt, 1977*; *Holt & Kotler, 1987*; *Holt & Lawton, 1994*). In these cases, interspecific aggression that leads to the spatial or temporal exclusion of prey or host species (i.e., individuals of the subordinate species) could reduce the overall density of prey or hosts, and thus reduce predation or infection rates of the dominant species (*Martin & Martin, 2001b*). Other hypotheses proposed to explain aggressive interactions among species include misplaced aggression (*Murray, 1976*; *Murray, 1981*; *Murray, 1988*; *Jones et al., 2016*), sexual selection for aggressive displays (*Nuechterlein & Storer, 1985*), and practice for intraspecific contests (*Nuechterlein & Storer, 1985*); however, evidence to date suggests that many aggressive interactions reflect adaptive responses to reduce ecological costs for one or both species (*Robinson & Terborgh, 1995*; *Martin & Martin, 2001b*; *Leisler, 1988*; *Palomares & Caro, 1999*; *Peiman & Robinson, 2010*; *Blowes, Pratchett & Connolly, 2013*; *Losin et al., 2016*).

Given the ecological importance of aggressive interactions among closely related species, such interactions may have broad consequences for species assemblages and trait evolution (*Morse, 1974*; *Grether et al., 2009*; *Grether et al., 2013*; *Freshwater, Ghalambor & Martin, 2014*; *Martin & Ghalambor, 2014*). The nature of these consequences, however, depends on whether the outcome of aggressive interactions between species are symmetric, with both species regularly winning aggressive contests, or asymmetric, with one species winning the majority of aggressive contests. For example, if the outcomes of aggressive interactions are asymmetric, then selection may favor traits in the dominant species that enhance fighting abilities (*Young, 2003*; *Owen-Ashley & Butler, 2004*; *Donadio & Buskirk, 2006*) or that signal dominance to other species (*Dow, 1975*; *Flack, 1976*; *König, 1983*; *Snow & Snow, 1984*). Asymmetric interactions may also favor traits in the subordinate species that reduce the likelihood of heterospecific aggression, such as traits that reduce ecological overlap with dominant species (*Feinsinger, 1976*; *Willis & Oniki, 1978*; *Morse, 1974*; *König, 1983*) or that reduce aggression from dominant species (e.g., the loss of signals that induce aggression from the dominant species, or the evolution of signals that mimic the dominant or other dangerous species; *Gill, 1971*; *Feinsinger & Chaplin, 1975*; *Feinsinger & Colwell, 1978*; *Rainey & Grether, 2007*; *Prum & Samuelson, 2012*; *Prum, 2014*). Conversely, if aggressive interspecific interactions are typically symmetric, then selection may act similarly on the interacting species, potentially favoring individuals in both species that maintain exclusive territories (*Orians & Willson, 1964*; although interspecific territoriality can involve species with asymmetric relations as well). Thus, selection should shape the evolution of species' traits differently if aggressive interactions are symmetric versus asymmetric. Despite the importance of understanding the prevalence of asymmetric relationships among interacting species, relatively little is known about how common such patterns are in nature.

In this paper, we ask: how often are aggressive interactions among closely related species asymmetric? Although the outcomes of many aggressive contests among species are asymmetric (*Morse, 1974*; *Lawton & Hassell, 1981*; *Persson, 1985*), few studies have compared the frequency of asymmetric versus symmetric relationships between aggressively

interacting species. The studies that have examined this question have found asymmetric interactions to be common: (1) a study of 13 species of surgeonfish (Acanthuridae) on a barrier reef at Aldabra, Indian Ocean, found evidence for asymmetric interactions among 26 of the 27 species pairs that interacted aggressively (*Robertson & Gaines, 1986*); (2) a study of closely related species of birds in Amazonian Peru found that nine of the 12 focal species pairs exhibiting interspecific aggression also showed statistically significant asymmetries in their response to playback of heterospecific songs (*Robinson & Terborgh, 1995*); (3) a comparative study of interspecific killing among carnivorous mammals found asymmetric killing (i.e., only one species was known to kill the other, rather than both killing each other) in 18 of 19 species pairs that were within the same taxonomic families (excluding domesticated species; *Palomares & Caro, 1999*); and (4) our own comparative study of ecological traits of dominant and subordinate species of North American birds found evidence that 64 of 65 congeneric species pairs had asymmetric outcomes to aggressive interactions (*Freshwater, Ghalambor & Martin, 2014*).

Here, we compile published, quantitative data on the outcomes of aggressive interactions among species within the same taxonomic families, focusing on birds where interaction data are common. We estimate asymmetries in interactions among species using statistical tests for asymmetries and the proportion of aggressive contests won by each species. Although statistical tests provide an accepted method for identifying asymmetries in the outcomes of interactions (*Crawley, 2013*), these tests may not be the optimal method for estimating their magnitude, prevalence, or biological importance. For example, a lack of statistically significant dominance asymmetries may simply reflect small sample sizes; in other cases, large sample sizes may result in statistically significant asymmetries of small biological effect (e.g., 55:45 splits in the outcomes of aggressive interactions). Thus, we also estimated the prevalence of asymmetric interactions by calculating the proportion of aggressive contests won by each species, providing a view of their biological importance that has not been highlighted in other studies to date (e.g., *Freshwater, Ghalambor & Martin, 2014*). The outcome of aggressive contests, including which species is behaviorally dominant, may also vary across different habitats or geographic locales (*Altshuler, 2006*; *Carstensen et al., 2011*), but the frequency of such variation has not previously been explored. To test how common asymmetric aggressive interactions are in birds, we examined the outcome of contests across diverse groups of birds, including vultures feeding at carcasses, hummingbirds feeding at nectar sources, antbirds and woodcreepers feeding on prey fleeing from army ant swarms, and a broad collection of North American congeners. Where possible, we also examined if the outcome of aggressive interactions between the same species pairs changed between different geographic locations. Collectively, we present results from data representing 270 interacting pairs of species from 26 families, and including the outcomes of 26,212 interactions.

## MATERIALS & METHODS

### Interaction data

We used published data from *Freshwater, Ghalambor & Martin (2014)* and *Martin & Ghalambor (2014)*, supplemented with additional quantitative data, including published

data for interactions that had been excluded from *Martin & Ghalambor (2014)* because of a lack of genetic or mass data for the interacting species. This study did not require vertebrate ethics approvals because we used published data in a comparative test supplemented with a few additional natural history observations. For data on North American congeners, we included only the youngest phylogenetically-independent species pair for which we had quantitative data on the outcomes of aggressive interactions (*Freshwater, Ghalambor & Martin, 2014*). We note that examining only the youngest phylogenetically-independent species pairs was important in our previous work that focused on the evolution of traits associated with dominance status (*Freshwater, Ghalambor & Martin, 2014*), but was not part of our approach to addressing the focal questions of this study. The complete datasets and sources for all of the data are included as Data S1–S3. Overall, we created two different datasets: (1) all of the data combined, including data for species interactions that were gathered from multiple sites and summed together for each species pair (Data S2); and (2) the same data entered for each individual location separately, and where each location had at least six observations per species pair (Data S3). We included data separated by location to address the potential effects of lumping data across geographic locations on our results. Separating data by location also allowed us to test for geographic variation in dominance relationships among species using the cases where the same species pairs had interaction data from multiple locations. For all datasets, we included only species pairs (Data S2) or locations (Data S3) that had at least six interactions with clear outcomes (i.e., one species clearly won the interaction). We rarely had data on the number of individuals involved across interactions, in part because most studies of aggressive interactions did not involve marked individuals. Thus, for some of the species pairs, the number of interactions includes some degree of pseudoreplication (where the same individual was involved in multiple aggressive interactions), but the extent of pseudoreplication across our dataset is unknown. Following the previous work, we included chases, supplants and displacements, kleptoparasitism, and physical attacks as aggressive interactions (see *Freshwater, Ghalambor & Martin, 2014* for definitions of these terms). We excluded observations that involved the defense of eggs or young and avoided interactions involving more than one individual of each species (following *Freshwater, Ghalambor & Martin, 2014*; *Martin & Ghalambor, 2014*). We included observations related to competition for nest sites, because many birds compete aggressively with other species for nesting sites (e.g., *Wallace, Collier & Sydeman, 1992*).

## Statistical tests of asymmetry

We tested for asymmetries in the outcomes of aggressive contests between pairs of species using binomial tests in the statistical program R (*R Core Team, 2014*). We ran binomial tests on aggressive interaction data for each species pair in our analysis, and again on our dataset partitioned by location within each species pair. The likelihood of detecting a significant ($P < 0.05$) asymmetry in the outcome of aggressive interactions among species increases with the number of interactions observed (i.e., sample size; *Crawley, 2013*), so we plotted $P$-values as a function of sample size for all species pairs.
## Proportion of interactions won

We also tested for asymmetries in the outcomes of aggressive contests between pairs of species by examining the proportion of interactions won by one species. We know of no cut-off for designating interactions as asymmetric, so we plotted the cumulative number of species pairs showing asymmetric outcomes to their interactions, varying the definition of asymmetric from >60% to 100% of the interactions won by the dominant. As before, we plotted these relationships for data summarized by species pairs, and again for data partitioned by location within each species pair.

## Variation in dominance among locations

For species pairs with dominance data from multiple locations (each location with greater than six interactions per species pair), we looked at the frequency with which dominance status switched between species among locations, and tested for differences in the proportion of aggressive contests won by each species between sites using Chi-squared tests in R (*R Core Team, 2014*). We estimated the distance between different geographic locations for each species pair by recording the latitude and longitude in decimal degrees for each site (from the original references, or estimated based on the description of the site within the original references), and then calculating the distance between these points in km using the *deg.dist* function in the R package *fossil* (*Vavrek, 2012*).

# RESULTS

## Statistical tests of asymmetry

Overall, 224 of 270 species pairs (83.0%) showed statistically significant ($P < 0.05$) asymmetries in the outcomes of aggressive contests. Above a sample size of 52, all aggressive interactions among species were statistically significant ($P < 0.05$) (Fig. 1). Data partitioned by location within species pairs revealed similar results: 235 of 288 comparisons (81.6%) showed statistically significant ($P < 0.05$) asymmetries in the outcomes of aggressive contests.

## Proportion of interactions won

For data summarized by species pair, 79.6% of species pairs had dominant species that won ≥80% of the aggressive contests (range across groups: 72.2% for vultures to 86.2% for antbirds and woodcreepers; Fig. 2). In contrast, 97.0% of species pairs had dominant species that won ≥60% of the aggressive contests (range across groups: 95.6% for hummingbirds to 100.0% for vultures), while 48.1% of species pairs had dominant species that won 100% of the aggressive contests (range across groups: 27.8% for vultures to 64.6% for antbirds and woodcreepers) (Fig. 2). Data partitioned by location within species pairs revealed similar results.

## Variation in dominance among locations

Across all species pairs, 28 had aggressive interaction data from more than one location (with over six interactions observed from each location); 21 species pairs had data for two locations, seven species pairs had data for three locations. The average distance among

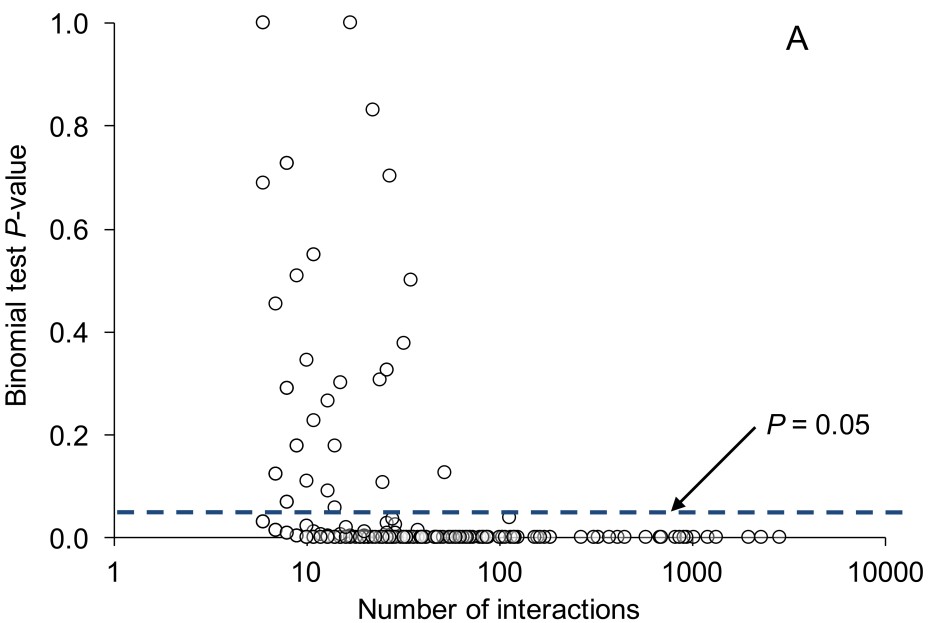

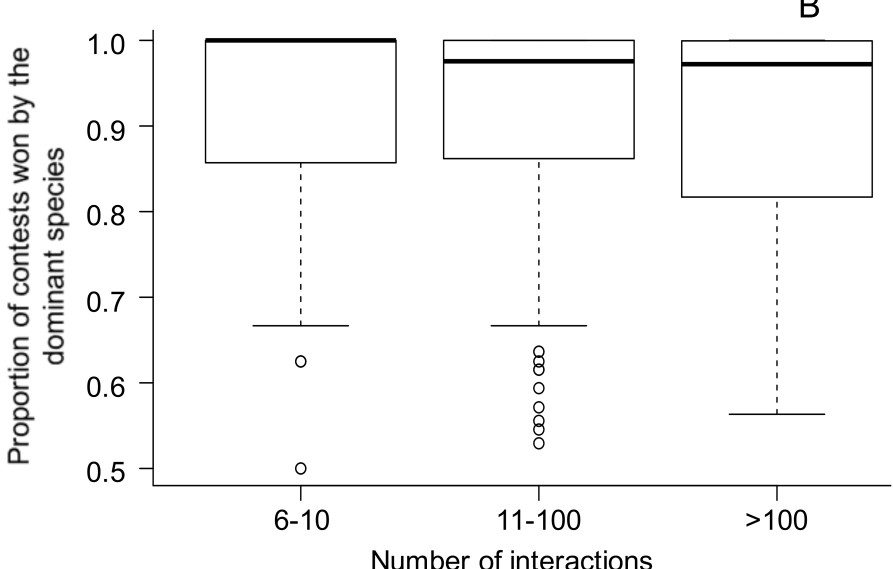

**Figure 1** **Relationship between the number of aggressive interactions observed between each species pair and (A) binomial test *P*-values testing for asymmetries in the outcomes of aggressive interactions, and (B) the proportion of aggressive contests won by the dominant species.** In (A), the dashed line illustrates the common *P*-value cutoff for statistical significance at 0.05. All species pairs with greater than 52 interactions showed statistically significant asymmetries; overall, 83% of species pairs showed statistically significant asymmetries. In (B), box plots show the median as a center line, the interquartile range as a box, values within 1.5*interquartile range as whiskers, and all data that lie outside the whiskers as circles. Overall, 84 species pairs had sample sizes between 6–10 interactions, 151 species pairs had 11–100 interactions, and 35 species pairs had >100 interactions.

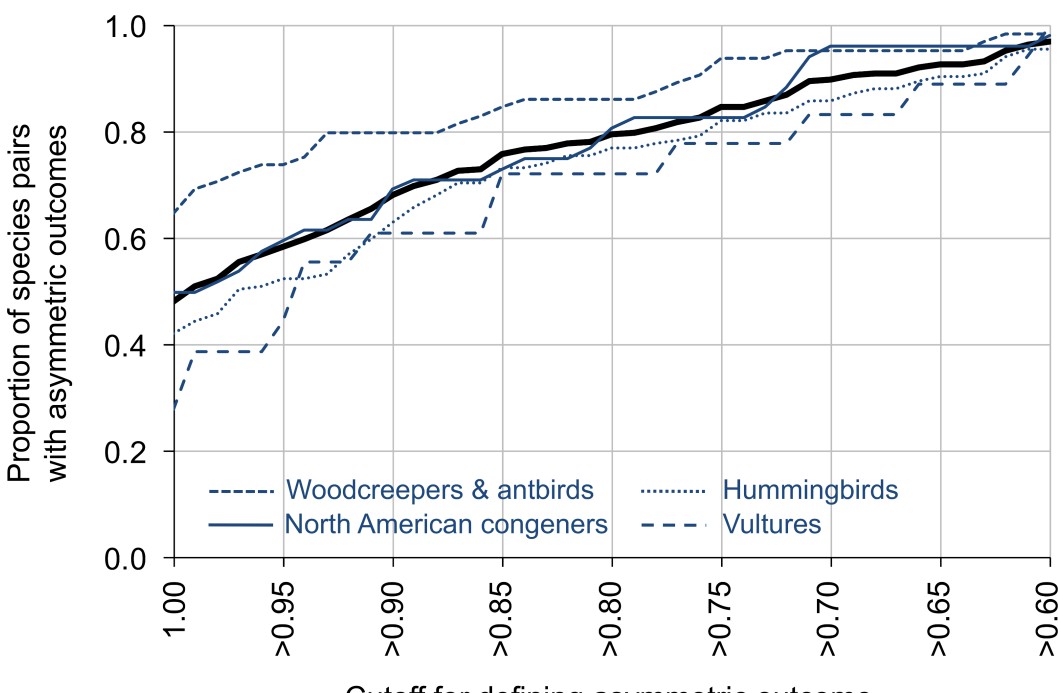

**Figure 2** **The proportion of species pairs showing asymmetric outcomes to their aggressive interactions.** Asymmetry was measured by the proportion of interactions won by the dominant species, and was defined on a scale from >60% of the interactions won by the dominant species to 100% of the interactions won by the dominant species ($x$-axis). The thick black line represents the entire dataset; the blue lines represent different groups within the dataset. Plots are line plots connecting points at 0.01 $x$-value increments. The sample sizes for the different groups are: vultures ($N = 18$ comparisons, 5,820 interactions), hummingbirds ($N = 135$ comparisons, 6,685 interactions), woodcreepers and antbirds ($N = 65$ comparisons, 9,263 interactions), North American congeners ($N = 52$ comparisons, 4,444 interactions).

geographic locations within species pairs was 1,176 km (range 13–4,184 km; all distance data are available in Data S4).

Dominance relationships within species pairs were consistent across sites (i.e., the same species won the majority of the interactions at both or all three locations) in 26 of the 28 species pairs (92.9%). The two species pairs whose dominance relationship flipped between locations included one pair of vultures (Accipitridae: Rüppell's Vulture, *Gyps rueppellii*—White-backed Vulture, *Gyps africanus*, Amboseli National Park, Kenya and Serengeti National Park, Tanzania) and one pair of hummingbirds (Trochilidae: Glittering-bellied Emerald, *Chlorostilbon lucidus*—Ruby-topaz Hummingbird, *Chrysolampis mosquitus*, Serra do Pará, Pernambuco, Brazil and Cadeia do Espinhaço, Bahia, Brazil). For five of the 28 species pairs (17.9%; including the two for which dominance relationships flipped between sites), the proportions of aggressive contests won by each species were significantly different among locations (i.e., Chi-squared test, $P < 0.05$). For the remaining 23 species pairs (82.1%), the proportion of aggressive contests won by each species did not differ significantly across sites.

In addition to consistent dominance relationships among locations, species pairs also showed the same dominance relationships in winter and summer ($N = 2$ migrant species pairs; *Bucephala islandica—clangula*; *Anas strepera—americana*) and in captive versus wild populations ($N = 3$ species pairs; *Ammodramus maritimus—caudacutus*; *Melospiza melodia—georgiana*; *Spinus psaltria—lawrencei*), suggesting that such asymmetries are repeatable across different contexts.

## DISCUSSION

Whether the outcome of aggressive interactions is commonly symmetric or asymmetric has important ecological and evolutionary implications. We found the outcomes of most aggressive interactions among species within the same taxonomic bird family were asymmetric. Overall, 83% of the 270 species pairs showed statistically significant asymmetries in the outcome of aggressive contests (i.e., binomial tests, $P < 0.05$; Fig. 1), with all species pairs showing statistically significant asymmetries above a sample size of 52 interactions. When we characterized asymmetry using the proportion of interactions won by the dominant species, we found that 80% of the species pairs contained dominant species that won 80% or more of aggressive contests (Fig. 2). For 28 species pairs, we had dominance data for 2 or 3 different populations, allowing us to test whether dominance asymmetries among species were consistent across locations. Dominance relationships were the same across locations for 93% of the species pairs (i.e., the same species won the majority of aggressive contests across the different locations), while the proportion of interactions won by each species was not significantly different across locations for 82% of the species pairs. These results suggest that dominance relationships between species are usually consistent across different sites. Below, we discuss the ecological and evolutionary consequences of asymmetric interactions, the factors that underlie dominance, the possible reasons to explain the few cases where dominance differed across different locations, and the implications for how dominant and subordinate species respond to human impacts.

### Asymmetric interactions and their consequences for ecology

Asymmetric outcomes to most aggressive interactions suggest that dominant species can use preferred resources and reduce the access of subordinate species to those resources (*Morse, 1974*). Such patterns are not unique to birds, as experiments have demonstrated asymmetric partitioning of resources in invertebrates (*Bovbjerg, 1970*; *Bertness, 1981a*; *Bertness, 1981b*) and across a diverse array of vertebrates (*Chappell, 1978*; *Robertson & Gaines, 1986*; *Alatalo & Moreno, 1987*; *Martin & Martin, 2001a*; *Hixon, 1980*; *Larson, 1980*; *Alatalo et al., 1985*; *Alatalo et al., 1987*; *Ziv et al., 1993*; *Pasch, Bolker & Phelps, 2013*). In these cases, subordinates are excluded from preferred resources, but are still able to use resources that cannot sustain dominant species, and thus are not preferentially defended (*Morse, 1974*; *Martin, 2014*). The result is a repeated pattern: dominant species direct aggression towards subordinate species (interference competition), restricting resource use for the subordinate species.

The strong asymmetries in the outcomes of aggressive contests also suggest that trade-offs involving aggressive ability and behavioral dominance could play an important role in the

partitioning of resources and coexistence of species, particularly among closely related species. For example, a smaller body size requires fewer resources to grow, survive, and reproduce, but comes at a cost in the form of losing aggressive contests to larger species (*Peters, 1983*; see also below). Given that resources vary in time and space, large- and small-sized species could coexist by partitioning habitats according to the abundance of resources, with aggressive interactions among species playing a central role in habitat partitioning (*Morse, 1974*; *Ford, 1979*; *Diamond et al., 1989*; *Martin, 2014*). We might expect other trade-offs involving aggressive abilities to lead to similar patterns of resource partitioning and coexistence among species of birds and other taxa (e.g., *Feinsinger, 1976*; *Willis & Oniki, 1978*; *Feinsinger & Colwell, 1978*).

## Asymmetric interactions and their consequences for evolution

Asymmetric aggressive interactions should represent an important selection pressure between interacting species (*Grether et al., 2009*; *Pfennig & Pfennig, 2012*; *Grether et al., 2013*). Our results suggest that we should expect such selection to favor the evolution of distinct traits and strategies that depend on the position of species within a dominance hierarchy (*Morse, 1974*; *Gauthreaux Jr, 1978*; *Grether et al., 2013*; *Freshwater, Ghalambor & Martin, 2014*). For example, selection may favor investment in aggression or territorial behavior in dominant species, even when such traits incur some fitness costs or trade-off with other traits. Selection may also favor traits, such as color patterns or displays, that signal dominance status to subordinate species in order to reduce the frequency and cost of aggressive encounters among species (see *Flack, 1976*; *König, 1983*; *Snow & Snow, 1984* for possible examples of these traits).

In contrast, selection might favor traits in subordinate species that reduce the costs of aggressive interactions with dominants. For example, the evolution of color patterns or displays in subordinate species may reduce aggression from dominant species, and thus be favored by natural selection (*Gill, 1971*; *Sætre, Král & Bičík, 1993*). The evolution of mimicry of dominant species by subordinates may also be an underappreciated outcome of asymmetric interactions among species. In birds, recent evidence suggests that the mimicry of dominant species by subordinates could be widespread, involving both song and color patterns (*Cody, 1973*; *Rainey & Grether, 2007*; *Prum & Samuelson, 2012*; *Prum, 2014*). Similarly, selection should favor traits that facilitate alternative ecological strategies that reduce the costs of aggressive contests with dominant species. Such alternative strategies could include altering the timing of breeding or geographic distribution to reduce temporal and spatial overlap with dominant species (*Freshwater, Ghalambor & Martin, 2014*), or evolving adaptations that allow subordinate species to use novel resources (e.g., physiological tolerance to conditions outside those experienced in a clade). With reduced access to important resources for survival, such as food and safe roosting sites, subordinate species might also be more likely to evolve distinct life history strategies that invest more in annual reproductive effort at the expense of annual survival (*Roff, 1992*; *Stearns, 1992*). Indeed, such patterns characterize dominant and subordinate species within a genus: subordinate species have lower annual survival rates and lay larger eggs for a given body mass (*Freshwater, Ghalambor & Martin, 2014*).

## What causes variation in the asymmetric outcomes of aggressive encounters among species?

Many factors have been identified as influencing the proportion of encounters won by dominant species, including age and sex (*Stiles, 1973*), as well as proximate factors that include condition, hunger level, density, and time of arrival or colonization (*Stiles, 1973*; *Lyon, 1976*; *Anderson & Horwitz, 1979*; *Cole, 1983*; *Wallace & Temple, 1987*; *Robinson, 1989*; *Tanner & Adler, 2009*). Perhaps the most important predictor of the outcome of aggressive contests, however, appears to be differences in body size among the interacting species (*Morse, 1974*; *Peters, 1983*; *Robinson & Terborgh, 1995*; *Donadio & Buskirk, 2006*; *Martin & Ghalambor, 2014*). Indeed, in the results we report here, the larger species was dominant in 87% of the contests where the outcomes of aggressive contests were asymmetric (defined as ≥80% of contests won by the dominant species), with the dominant species averaging 57% heavier than the subordinate (for a list of reasons why larger size confers an advantage in aggressive contests, see *Martin & Ghalambor, 2014*). This contrasts with cases where one species won between 50–69% of the contests (i.e., the outcome was more symmetric), where the larger species prevailed in only 67% of the species pairs and averaged only 25% heaver (data in Data S2).

The importance of body size for determining the outcomes of aggressive contests, however, can vary. For example, larger species win a greater proportion of aggressive interactions as the difference in body size between interacting species increases, but this relationship weakens with greater evolutionary distance among the interacting species (*Martin & Ghalambor, 2014*). We hypothesize that this pattern occurs because closely related species share more traits with each other (*Violle et al., 2011*), and thus differences in size alone can determine the outcome of aggressive interactions (*Martin & Ghalambor, 2014*). As species become more distantly related, however, they are more likely to accumulate unique traits that influence behavioral dominance independent of body size. Indeed, *Martin & Ghalambor (2014)* found that as species became more distantly related the outcome of aggressive interactions became more asymmetric independent of differences in body size. Few studies, however, have attempted to identify the exact suite of traits that explain dominance independent of body size (*Donadio & Buskirk, 2006*; *Martin & Ghalambor, 2014*).

## The rare flip: why does dominance shift for some species pairs across sites?

Although the outcomes of aggressive interactions were usually asymmetric and consistent across different locations, the dominant species differed across sites for two pairs of interacting species. We could not determine the cause of the variation in dominance among sites in either of these cases, although the two different vulture studies categorized the outcomes of aggressive interactions in different ways, which could have contributed to the different patterns (*Kruuk, 1967*; *Anderson & Horwitz, 1979*). Studies of other examples of dominance flipping, however, help to shed light on when and why such cases arise.

The most detailed study of shifting dominance across sites examined the relative fighting performance of hummingbirds at different elevations. Rufus Hummingbirds

(*Selasphorus rufus*) dominate Broad-tailed Hummingbirds (*S. platycercus*) at low elevations in Colorado, USA, but are subordinate at higher elevations (*Altshuler, 2006*). The dominance reversal across elevations appears to result from differences in wing loading between the species and how changes in air pressure alter flight performance (*Altshuler, 2006*). Specifically, at high elevations, the long-winged Broad-tailed Hummingbird appears better able to achieve burst power performance, and thus dominates Rufus Hummingbirds; at low elevations, burst power is unconstrained, and the greater maneuverability and sustained aerodynamic performance of the shorter-winged Rufus Hummingbird appears to provide a competitive advantage over Broad-tailed Hummingbirds in aggressive contests (*Altshuler, 2006*).

Similar variation in fighting abilities may also characterize species interactions in the air versus on the ground (or water), especially because a heavier weight improves fighting abilities on the ground, but can compromise aerial maneuverability that can influence the outcome of aggressive contests in the air (see *Peters, 1983*; *Bonner, 2006*). Such trade-offs in performance may explain different outcomes to aggressive contests for some species pairs (that were not part of our dataset in this paper); for example, Whistling Kites, *Haliastur sphenurus*, and European Herring Gulls, *Larus argentatus*, are dominant to Black Kites, *Milvus migrans*, and Audouin's Gulls, *Ichthyaetus audouinii*, respectively, on the ground, but are subordinate to them in the air (*Marchant & Higgins, 1993*: page 74; *Cramp, 1983*: page 784). Such trade-offs may be more widespread than is presently recognized, particularly in birds that commonly interact in both aerial and terrestrial contexts (e.g., Laridae, Accipitridae, Corvidae).

The relative densities of subordinate species can also influence the outcomes of aggressive contests, and cause variation in dominance across sites. In our study, interactions between Blue-throated (*Lampornis clemenciae*) and Magnificent (*Eugenes fulgens*) hummingbirds differed significantly between locations, with Blue-throateds dominant at two sites, but no significant difference in the number of contests won between the species at a third site. High densities of the subordinate Magnificent Hummingbird at the third site was thought to have created this shift; individuals of the dominant Blue-throated Hummingbird were more likely to abort and retreat from an aggressive interaction when they encountered high densities of subordinates, presumably because winning one interaction would be unlikely to give them sufficient access to the resource (*Lyon, 1976*). Thus, the density of subordinate species can influence the outcome of individual interactions, even with a lack of coordinated fighting among individuals (*Lyon, 1976*; *Martin & Ghalambor, 2014*). This influence of subordinate density is thought to explain variation in the outcomes of aggressive interactions across sites and contexts in vultures, hummingbirds, blackbirds, and perhaps other groups, where subordinate species show substantial variation in their densities (*Orians, 1961*; *Orians & Collier, 1963*; *Lyon, 1976*; *König, 1983*; *Wallace & Temple, 1987*; *Houston, 1988*; *Kirk, 1988*; *Buckley, 1996*).

Other cases of dominance flipping between sites may simply reflect changes in the composition or activities of populations over time or space, particularly with respect to age, sex, and territorial behavior. For example, adult male Anna's Hummingbirds (*Calypte anna*) dominated adult male Costa's Hummingbirds (*C. costae*) in 23 of 25 interactions

(92%); in contrast, immature Anna's Hummingbirds won only two of 20 interactions (10%) with adult male Costa's Hummingbirds (*Stiles, 1973*). The relative distributions of males and females, and adults and immatures, vary because sexes and age classes show different seasonal movements and distributions in many species (*King, Farner & Mewaldt, 1965*; *Myers, 1981*). In addition, territorial behavior and associated aggression often varies over time and space (*Nelson, Badura & Goldman, 1990*), potentially leading to variable levels of aggression across species at different times or in different sites. All of these factors, from the density of air to the territorial behavior of individuals can cause dominance relationships to flip between species at different sites. The rarity of this dominance flipping in our dataset, however, suggests that the most important determinants of dominance across species remain consistent regardless of geography, and despite the many other factors that can influence the outcomes of aggressive contests.

### Asymmetric interactions and their consequences for human impacts

Given repeated patterns of dominance and asymmetric interactions among species, we might expect species to differ in their responses to anthropogenic challenges, such as climate change and habitat alteration, depending on their position within a dominance hierarchy. Some subordinate species appear to be better able to persist in degraded habitats (*Daily & Ehrlich, 1994*), and may have greater ecological breadth and tolerance compared to dominant species (*Morse, 1974*; *Minot & Perrins, 1986*; *Blowes, Pratchett & Connolly, 2013*; but see *Freshwater, Ghalambor & Martin, 2014*). Aggression and behavioral dominance, however, are often associated with boldness that can help species cope in the face of human disturbance (*Evans, Boudreau & Hyman, 2010*; *Lowry, Lill & Wong, 2013*). Thus, traits that covary with dominance status could facilitate or hinder species in the face of human alteration of habitats. Regardless, the importance of asymmetric interactions in determining patterns of resource use among species suggests that these interactions may mediate species' responses to perturbations like habitat alteration or climate change. Few models consider these kinds of species interactions in their forecasts of the impacts of habitat perturbations or climate change on species abundance and distributions (*Tylianakis et al., 2008*; *Gilman et al., 2010*; *Buckley, 2013*). Yet, any impacts on dominant species are likely to have cascading effects on the subordinate species with which they interact (*Duckworth & Badyaev, 2007*; *Gilman et al., 2010*; *Jankowski, Robinson & Levey, 2010*; *Buckley, 2013*; *Martin & Dobbs, 2014*; *Freeman & Montgomery, 2015*; *Freeman, Class Freeman & Hochachka, 2016*). Such asymmetric interactions could have important consequences for populations, particularly in environments where the options for dispersal and range shifting are limited (e.g., tropical islands and mountains; *Jankowski, Robinson & Levey, 2010*; *Freeman, 2016*).

## ACKNOWLEDGEMENTS

We thank Fran Bonier for helpful insight and discussion, and Ben Freeman and Greg Grether for comments and suggestions that significantly improved the manuscript.

### Funding

Funding to support this work was provided by a Natural Sciences and Engineering Research Council of Canada grant to Paul R. Martin and a National Science Foundation grant IOS-1457383 to Cameron Ghalambor. The funders had no role in study design, data collection and analysis, decision to publish, or preparation of the manuscript.

### Grant Disclosures

The following grant information was disclosed by the authors:
Natural Sciences and Engineering Research Council of Canada.
National Science Foundation: IOS-1457383.

### Competing Interests

The authors declare there are no competing interests.

### Author Contributions

- Paul R. Martin conceived and designed the experiments, analyzed the data, wrote the paper, prepared figures and/or tables, reviewed drafts of the paper.
- Cameron Freshwater and Cameron K. Ghalambor conceived and designed the experiments, wrote the paper, reviewed drafts of the paper.

### Animal Ethics

The following information was supplied relating to ethical approvals (i.e., approving body and any reference numbers):

We used published data in a comparative test supplemented with a few additional natural history observations; this study did not require vertebrate ethics approvals.

### Data Availability

The raw data has been supplied as Supplemental Files.

### Supplemental Information

Supplemental information for this article can be found online at http://dx.doi.org/10.7717/peerj.2847#supplemental-information.

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
