# Peer review of "The outcomes of most aggressive interactions among closely related bird species are asymmetric"

_PeerJ, doi:10.7717/peerj.2847_

## Round 0.1 · original submission · Minor Revisions

Both reviewers found considerable merit in your paper and propose that you make some minor revisions to improve it. I look forward to reading a revised version.

·

Basic reporting

This paper is clearly written.

Experimental design

The design of this analysis is clear.

Validity of the findings

The findings of this study are valid.

Additional comments

In this ms, Martin, Freshwater and Ghalambor ask whether aggressive interactions between closely related bird species tend to be asymmetric. Their previous papers (and others) have shown that aggressive interactions between birds tend to be asymmetric, but it remains an open question how general this pattern is, and whether the symmetry of comparisons is consistent across space and time. The overall question is important because the evolutionary and ecological ramifications of symmetric species interactions are very different from when interactions are asymmetric. The authors find the clear result that asymmetric interactions prevail across all birds, and within 1) vultures, 2) hummingbirds, 3) ant-following sub oscines and 4) North American birds. This is a strong paper with a clear conclusion.

I have a small number of comments that I believe would further improve the paper:

line 135 - Can you comment on the potential for pseudoreplication in the underlying data? That is, it seems doubtful that each observed interaction in the dataset pertains to unique individuals. This is relevant because our confidence that species A is behaviorally dominant to species B is greater if we see ten unique individuals of A chase ten unique individuals of B than if we see the same individual of A chase the same individual of B ten times. The consistency of outcomes among locations suggests this is not a big issue, but it is worth noting this possible limitation of the dataset.

lines 205 - 212 - what new information is this providing? It is more confusing than helpful to me. The preceding paragraph nicely summarized the relationships when looking at each species-pair comparison, and the following paragraph nicely summarizes variation in species-pair comparisons among locations.

line 214 - It would be useful if the authors included information on geographic distance between multiple locations. I think the inference on the consistency of interactions is much stronger when locations are far apart (e.g., New World vultures, Egretta, etc) than if sites are quite close to one another (e.g., hummingbirds within Guanacaste, Costa Rica).

lines 223-229 - can you comment further on these examples of “flipped” relationships and what they might imply for coexistence and trade-offs? Though few in number, these still strike me as interesting.

Other comments relevant to the “partitioned by location” dataset.

- Lampornis and Eugenes almost flip between Arizona and Mexico.
- Bucephala and Anas seem to show the same relationship in both winter and summer. This is worth pointing out.
- Melanospiza seems to have the same relationship in captivity and the wild. This is also worth pointing out.
- There seems to be a more-or-less hierarchy in genera in the hummingbird data. For example, Boissonneaua is dominant over Heliodoxa regardless of the species identity. To the degree this is true, it is consistent with relationships changing minimally or not at all among locations.

line 369 - a pertinent recent paper is Freeman et al 2016 Ibis 158: 726-737

Figure 1 - A lot of this relationship is driven simply by how binomial tests work. It is not surprising that cases with high sample sizes show significant p-values — if the dominant species wins 66% of the time this is not significant until there are 50 observations. And if the dominant species wins 50% of the time this is not significant until there are 100 observations. The point is that it gets harder and harder to NOT show a significant asymmetry with large sample sizes, to the point that a 53 vs 47 split is significant with just over 1000 observations. So I don’t find the statistical significance of asymmetry all that compelling. A histogram of % contests won by the dominant species with each species-pair would be more helpful, or perhaps multiple histograms that show the relationships for different levels of data quality (one for cases where the total number of observations is less than 20, one where the total number of observations is 20 - 100, etc).

Figure 2 - Could the authors add the sample size of species-pair comparisons for each of the four groups, either in the figure or the legend.

·

Basic reporting

I reviewed an earlier draft of this manuscript for a different journal and all of my comments on the previous draft are thoroughly addressed in the current draft. As I said previously, I think this is a valuable contribution.

In general, the manuscript is well written. A few sentences in the Introduction need work:

Lines 50-53: This sentence is hard to follow and includes a typo. If this is an important point, I think it should be clarified further.

Lines 55-62. Also hard to follow and includes a typo.

Line 83. This might be interpreted to mean that asymmetry in aggressive interactions precludes interspecific territoriality. There are many cases of interspecific territoriality in which one species dominates the other. Also, interspecific territoriality is a species interaction, not a trait.

This manuscript is based mainly on a reanalysis of data that were published elsewhere but with one exception (mentioned in the next section) it is self contained and could be understood without knowledge of the authors' previous papers.

I have not studied the PeerJ template but the manuscript conforms to standard formatting for scientific articles.

Experimental design

This is not experimental study, but the design of the study is sound. For the most part, the methods are described clearly and seem adequate. As I understand them, the criteria used for including/excluding data seem appropriate. I just have one concern along these lines:

Lines 140-143: What was the rationale for including only the youngest phylogenetically independent species pair for N. Am. congeners? And what does this mean, exactly? Based on this statement, I was expecting to find each species included in only one species pair in the dataset, but that's not the case. Whatever was done, why was it only done for N. Am. but apparently not elsewhere?

Validity of the findings

The basic conclusions are that most aggressive interactions between birds of different species are asymmetric, that the probability of detecting an asymmetry increases with the number of observations of such interactions, and that interactions between species that differ more in body size tend to be more asymmetric. All of these conclusions follow from the results presented.

The discussion does not range too far from the results presented and in general makes sense and makes good use of the literature. One question:

Lines 267-269: Are indirect interactions part of the pattern being described here? If so, this requires more explanation. If not, I don't think this belongs in the sentence.

The data are included as a supplement and easy to understand.

---

## Round 0.2 · accepted · Accept

I have read your responses to the reviewers' comments and am satified with the current state of the manuscript. Nicely done.